# Self-Administration of Long-Acting Somatostatin Analogues in NET Patients—Does It Affect the Clinical Outcome?

**DOI:** 10.3390/medicina57121287

**Published:** 2021-11-23

**Authors:** Anna Sowa-Staszczak, Marta Opalińska, Anna Kurzyńska, Karolina Morawiec-Sławek, Aleksandra Gilis-Januszewska, Joanna Palen-Tytko, Helena Olearska, Alicja Hubalewska-Dydejczyk

**Affiliations:** 1Department of Endocrinology, Jagiellonian University Medical College, 31-008 Kraków, Poland; anna.sowa-staszczak@uj.edu.pl (A.S.-S.); a.kurzynska@uj.edu.pl (A.K.); karolina.morawiec-slawek@uj.edu.pl (K.M.-S.); myjanusz@cyf-kr.edu.pl (A.G.-J.); h.zwinczewska@doctoral.uj.edu.pl (H.O.); alahub@cm-uj.krakow.pl (A.H.-D.); 2Nuclear Medicine Unit, Department of Endocrinology, Oncological Endocrinology and Nuclear Medicine, University Hospital, ul. Jakubowskiego 2, 30-688 Kraków, Poland; 3Department of Endocrinology, Department of Endocrinology, Oncological Endocrinology and Nuclear Medicine, University Hospital, 30-688 Kraków, Poland; jpalen@su.krakow.pl

**Keywords:** lanreotide, octreotide, self-administration, neuroendocrine tumors, treatment outcome

## Abstract

*Background and Objectives*: Long-acting somatostatin analogues (SSA) (octreotide LAR and lanreotide Autogel) are recommended as first line treatment of locally advanced or metastatic well-differentiated neuroendocrine tumors (NETs) with a good expression of somatostatin receptor (SSTR). Both of these SSAs are usually administered via injections repeated every 4 weeks. The purpose of the study was to compare the route of SSA administration (injection performed by professional medical staff and self-administration of the drug) with progression-free survival. *Materials and methods*: 88 patients in 2019 and 96 patients in 2020 with locally advanced or metastatic well-differentiated NETs were included in the study. All patients had a good expression of SSTR type 2 and had been treated for at least 3 months with a stable dose of long-acting somatostatin analogue every 4 weeks. All of them had received training on drug self-injections from professional NET nurses at the beginning of the COVID-19 epidemic. *Results*: The rate of NET progression in the study group in 2020 was higher than in 2019 29.1% vs. 18.1% (28 vs. 16 cases), *p* = 0.081. *Conclusions*: The method of administration of long-acting SSA injection performed by professional medical staff vs. self-injection of the drug may significantly affect the risk of NET progression. The unequivocal confirmation of such a relationship requires further observation.

## 1. Background

The introduction of somatostatin analogues (SSA) was a major therapeutic advance in the management of well-differentiated neuroendocrine tumors with good somatostatin receptors type 2 (SSTR2) expression. Long-acting formulations, usually administered every 4 weeks, have an antiproliferative effect, confirmed in clinical trials [1,2], and were proven to effective in decreasing serum levels of different hormones in the case of hormonally active neuroendocrine tumors (NETs) [3]. It can reduce the severity of clinical symptoms in most patients. Currently, two drugs are registered for the treatment of NETs patients: lanreotide Autogel for deep sub cutaneous injection and octreotide long-acting release (LAR) for intra muscular injection. It was shown that an intramuscular injection of octreotide, even done by nurses, encounters difficulties in proper administration of the drug, which translates into worse control of the carcinoid syndrome [4]. In the case of lanreotide drug administration, subcutaneous vs. intramuscular injections revealed similar drug efficacy, and now the subcutaneous injection is a standard clinical practice [5].

The availability of a ready-to-use preparation (in the case of lanreotide Autogel) resulted in the possibility of performing an at-home self-administration of the drug by patients or other trusted persons. However, such an option was not a common choice in the case of NET patients at our center, mainly due to administrative reasons and lack of patients’ interest. According to registration materials, in the case of octreotide LAR, injection should be performed by qualified staff. In both cases the injection tolerance is good, but according to the literature the administration of lanreotide Autogel is preferred by medical staff due to a more convenient syringe [6]. In theory, self-administration of drugs could be beneficial to patients, mainly due to the fact it would spare the time needed for medical visits, reduce the impact of the disease on patients’ daily life and improve the acceptance of long-term treatment. Furthermore, this kind of drug administration is currently successfully used in the treatment of acromegaly [7], but due to drug reimbursement regulation it is not widely used for NETs patients.

Restrictions and recommendations of the government related to the COVID-19 pandemic have led to a reduction in the number of medical visits, including those related to chronic treatment, even in oncology.

The aim of the study was to determine whether route of SSA administration (via an injection performed by professional medical staff before the epidemic, self-administration of drugs performed by patients or an injection performed by a nurse at General Practitioner’s office), forced by the restrictions related to COVID-19 pandemic, has an impact on the efficacy of treatment in NETs patients.

## 2. Materials and Methods

The study involved 184 patient-years observation with disseminated or inoperable well-differentiated NETs, treated at the Department of Endocrinology in Krakow, Poland (88 patients treated between April 2019 to January 2020 and 96 (72 continuing treatment from first group and new cases) treated between April 2020 to January 2021, named 2019 and 2020 group respectively). All patients were diagnosed with locally advanced or metastatic well-differentiated NET with good expression of SSTR type 2, and had been treated for > or = 3 months with a stable dose of long-acting somatostatin analogue (lanreotide Autogel 120 mg or 30 mg of octreotide LAR every 4 weeks according to the current guidelines). Since April 2020, due to a change in the drug-regulating law, forced by the SARS-CoV-2 epidemic, monthly SSA injections at a specialist medical center were replaced with self-injections of SSA (performed either by the patients themselves or at General Practitioner’s office). All patients had received training on self-injection of drugs from professional NET nurses. On every on-site visit the interview regarding difficulties in self-injection of the drug and the severity of clinical symptoms was performed and consulted if needed. A retrospective analysis was performed on the basis of patients’ medical documentation. Time to progression was defined as the time from the administration of the first SSA dose to the time of progression confirmed by radiological examinations: computed tomography (CT) or magnetic resonance imaging (MRI). In some cases, functional imaging–somatostatin receptor imaging (SRI) with Ga-68 labelled somatostatin analogue was also performed. The patients who progressed were excluded from further follow-up. The descriptive statistics were provided as mean, range and percentage values. The chi-square test was used to assess the differences in the incidence of progression in group of patients in 2019 and 2020.The other differences between groups were calculated using the chi-square test, exact Fisher test, unpaired *t*-tests and the Mann–Whitney-test as indicated with the use of IBM SPSS Statistics for Windows, version 26 (IBM Corp., Armonk, NY, USA). The results were considered to be statistically significant with *p* < 0.05.

## 3. Results

The total number of patients treated with long-acting SSA due to disseminated or inoperable NET (G1 or G2) was 88 in 2019 (since April) and 96 (72 continuing treatment from first group) in the consecutive period in the year 2020. Among these patients, there were 16 cases (18.1%) of progressive disease (PD) (11 women, 10 men) in 2019, while in 2020 there were 28 patients (29.1%) with PD (13 women, 14 men) (*p* = 0.081). The mean time to progression was 57.9 months (range 8–144) in 2019, and 55.18 months (range 6–152) in 2020. The analyzed groups did not differ in terms of age, the follow-up period, the performance status and the tumor burden (assessed as liver, lymph nodes and bone involvement) (Table 1). In 2019, progression was not observed in patients with G1 tumors, while in 2020 it was observed in 6 patients with G1 (*p* = 0.013).

The greatest increase in the incidence of progression was seen among patients with small intestine NETs. There was not statistically significant difference between groups receiving Sandostatin LAR and Somatuline Autogel in consecutive years. The numbers of patients with PD in 2019 and 2020 independent of the primary focus localization are presented in Figure 1. 

## 4. Discussion

Self-administration of various drugs via injection has been widely known for many years and is routinely used in the treatment of many diseases.

There is a growing body of literature recognizing the importance of this approach; so far, however, self-administration of drugs has not been widely used in the case of NETs patients. Due to the COVID-19 pandemic, the number of visits at outpatient clinics had to be reduced, with at-home treatment being introduced wherever possible. It coincided with changes in the existing medical procedures in Poland. As a result of the latter point, all patients who had accepted this kind of treatment could receive an SSA analogue at home for a period of 3 months, performing self-injection at home or having it performed at a General Practitioner’s office, with follow-up visits at our NET center every 3 months.

So far, only two SSAs have been approved as first or second-line treatment of disseminated or inoperable well-differentiated neuroendocrine tumors with good SSTR expression or hormone production. Although they both have similar medical properties (antiproliferative and antisecretory effect in NETs [8]). A systematic literature review presented that majority of patient favor use of lanreotide Autogel over octreotide LAR [9]. There are several practical differences regarding the drug formulation and the route of administration [10]. Lanreotide Autogel is provided in the form of a prefilled syringe for self-administration, which consists in performing a deep, subcutaneous injection, while octreotide is provided in the form of powder for reconstitution, for an intramuscular injection, and should be administrated by professional medical staff. According to previous publications, patients’ general health-related quality of life (HRQoL) (evaluated using SF-12) was similar in two treatment groups, moderate-to-severe anxiety before injections (analyzed independently to HRQoL) was reported more frequently by patients treated with octreotide (11%) than those treated with lanreotide (2%) [11], but none of the studies compared the head-to-head oncological efficacy of the treatment. 

In the case of evenly disseminated NETs patients, progression-free survival (PFS) after initial treatment with SSA remains relatively long [8], which further encourages the consideration of treatment involving self-injection. Even though the self-administration of drugs seems to be an attractive treatment option for patients, some concerns are very important and need to be validated in real-life settings.

Before the decision regarding the change in the treatment method was made, all patients at our center has been provided with training on self-injection of SSA by NET-experienced nurses; the training was repeated as many times as necessary. According to data collected directly from patients, most of them decided to perform the administration of SSA on their own. The most common reason for deciding not to perform self-injections were the lack of support or reluctance to receive SSA in a General Practitioner’s office, which probably stemmed from a relatively high price of the drug and the lack of possibility to receive a training on SSA injections at an experienced NET center, especially with regard to octreotide. 

A key question regarding self-injection of long-acting SSA is how to assess patients’ actual adherence to recommendations, however the problem associated with unpleasant sensations connected to the injections does not appear to be clinically relevant. In survey study inspecting patients’ satisfaction with long-acting injectable somatostatin analog therapy for neuroendocrine tumors significant percentage of patients(70.2% (*n* = 87 of 124) reported mild to no pain or discomfort at the injection at 28 day post-injection [12]. 

The second problem is the determination of whether self-injection is performed with the use of a correct technique. A study of 115 patients evaluating the results of 328 gluteal octreotide injections showed successful rate of drug administration in only 52% [4]. Further repeated injections at the same sites may be cause decreasing octreotide efficacy due to granulomatous reactions [13]. In our cohort the difference in progression risk in the case of both somatostatin analogue was not statistically significant despite different method of injection (intramuscular vs. subcutaneous). Finally, since both SSAs must be transported and stored in a fridge, it cannot be checked before the injection whether quality of the drug is maintained. Only proper quality drug ensures good results of the treatment. Faultily octreotide LAR preparation may change the size, distribution or thickness of the microsphere’s polymer coating which can lead to significant change of drug-release characteristics [13]. However all of these factors may be improved by adequate training of patients.

The increase in the progression rate observed in our study that occurred in a relatively short time is disturbing, despite the fact that the results did not reach statistical significance (*p* = 0.081). It is worth noting that at the time of self-injection we observed an increasing ratio of progression, especially in patients with G1 tumors, potentially in the group in which we could expect a relatively long period of disease stabilization. 

As described in the literature, SSTR2 desensitization and lowering of their suppression during octreotide and lanreotide treatment, particularly in GEP-NETs, were revealed in clinical observation [14]. That mechanism seems to be less probable in a such short observational period than the incorrect drug administration. Moreover, such progression rate was not seen in the previous period.

Additionally, although not significant, a higher progression ratio was seen in the time of self-administration in the group of patients with a lower Karnofsky status, which may be connected with lower compliance rates due to disability.

Looking at the cross-section of patients, good access to training, the most likely reason for increasing the number of patients with progression seems to be incorrect drug injection technique, deliberate or random skipping of injections or incorrect drug storage (given to patients for every 3 months for self-storage). Nevertheless, asked about problems with self-injection, patients rarely report difficulties or admit missing doses of the drug nor exacerbation of clinical symptoms regardless of which type of somatostatin analogue they received. This observations in our report may indicate a failure of the treatment system and could suggest that not all patients are good candidates for such treatment protocol. First of all, we should try to improve the identification of patients who are capable of performing self-injections and consider proper training not only for patients, but also for nurses working at General Practitioner’s offices. Perhaps an active suggestion about possibility of the consultation with professional NET nurses should be addressed to GP-nurses and patients family members in the case of any problems with drug administration. Further injection monitoring systems based on information about injections obtained from the patients could be useful for improving the quality of the treatment. A survey prepared for implementation in our center is presented in Table 2.

The potential value of this study stems from the possibility of comparing two clinical situations, in which similar (NET G1 and G2) patients were receiving drugs under the supervision of a qualified nurse and performing self-injections on their own.

The main limitation of this study was the relatively short evaluation time, which comprised of only two selected short observation periods, resulting in a risk of study bias; therefore, the applicability of the results remains uncertain. Moreover, the results of the statistical analysis are on the verge of statistical significance, which could stem from a relatively small number of patients, recruited from only one center. With regard to NET progression, it is also necessary to take into consideration other prognostic factors, such as age, tumor burden, the general condition of a patients or the type of the SSA used. However, the suspicion of a significant reduction in the effectiveness of the applied treatment requires attention and comparison of data collected from other centers. 

## 5. Conclusions

Most patients with well-differentiated NETs or their partners were able to administer SSA injections outside of outpatient (NET experienced) clinics (at home or at General Practitioner’s office). Unfortunately, unsupervised at-home injections were associated with a higher rate of PD during a one-year observation period, compared to injections performed by professionals (at an experienced NET center).

## Figures and Tables

**Figure 1 medicina-57-01287-f001:**
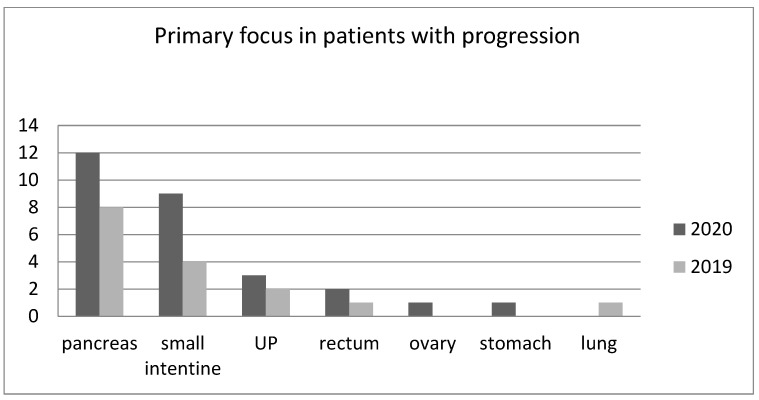
The numbers of patients with progressive disease in dependence on the location of the primary lesion in 2019 and 2020 (UP—unknown primary site of NET).

**Table 1 medicina-57-01287-t001:** Comparison of patient groups with disease progression in 2019 and 2020.

Variables	Patients with Progression in 2019 (no. 16)	Patients with Progression in 2020 (no. 28)	*p*-Value
Mean age in years (range)	62.50 (46–79)	62.42 (42–79)	0.56
Mean follow-up period in months (range)	57.88 months (8–144)	55.18 months (6–152)	0.724
Grading, no. (%)			0.013
G1	0%	6 (21.4%)	
G2	12 (75.0%)	20 (71.4%)	
G—not established	4 (25.0%)	2 (7.1%)	
Performance status (Karnofsky scale)			0.79
≥80	14 (87.5%)	23 (82.1%)	
<80	2 (12.50%)	5 (17.9%)	
Distant NET metastases at the beginning of observational, period, no. (%) of patients with:			0.798
Liver involvement	15 (93.8%)	26 (92.3%)	
Lymph node involvement	11 (68.8%)	17 (60.7%)	
Bone involvement	8 (50.0%)	12 (42.9%)	
Type of somatostatin analogue			0.497
Sandostatin LAR	6 (37.5%)	12 (42.9%)	
Somatuline Autogel	10 (62.5%)	16 (57.1%)	

**Table 2 medicina-57-01287-t002:** A survey proposed for improvement and validation of the quality of the self-injections.

Injection Monitoring System—Card No.
Phone number to your NET experienced nurse:
Patient’s first and last name:
Name of somatostatin analogue:
Dose of somatostatin analogue:
Planed date of injection:
Was the medicine transported in a refrigerator?
☆ yes ☆ no.
Was the medicine stored in a refrigerator?
☆ yes ☆ no.
Date of planned injection:
Type of injection:
☆ Self-injection ☆ Injection given by friend/family member ☆ Injection given by GP-nurse
Place of injection:
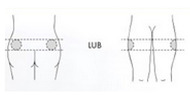
Problems concerning:
Drug preparation Injection Side-effects after injection

☆—please tick the appropriate

## Data Availability

The data presented in this study are available on request from the corresponding author.

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
