# Peer review of "Self-Administration of Long-Acting Somatostatin Analogues in NET Patients—Does It Affect the Clinical Outcome?"

_medicina, 2021, doi:10.3390/medicina57121287_

Round 1
Reviewer 1 Report
The topic is interesting and important
Authors have reported the study in a very understandable way but still some aspects were not discussed
- Author didn't address the scientific and clinical reasons that led to their results. Only they attributed the results to the injection technique and drug storage.
- The study is poor of references.
- No up to date references although there are many studies are already published such as this recent work "Darden, C., Price, M., Ray, D. et al. Patients’ satisfaction with long-acting injectable somatostatin analog therapy for neuroendocrine tumors. J Patient Rep Outcomes 5, 82 (2021). https://doi.org/10.1186/s41687-021-00355-5"
- The study has to be more deep.
Author Response
Thank you very much for your comments concerning the article „ Self-administration of long-acting somatostatin analogues in NET patients – does it affect the clinical outcome?”. Following your suggestions we corrected the text closely to your comments.
- The discussion were extended and some clinically relevant information were added.
- The list of reference was updated.
- The mentioned reference was added to discussion.
- In addition to extending of the discussion, we have added a proposal of Injection monitoring system card, which we plan to implement in our center and which we hope, will have a positive impact on patients compliance and their satisfaction with treatment. Please see the attachment

Reviewer 2 Report
General considerations
It is a well-written manuscript on a current and relevant subject. This manuscript needs some corrections and some clarifications for its better understanding.
Specific Considerations
Please check that the terms in the Keywords are contained or matched in the Medical Subject Headings (MeSH).
Would you please correct the Student’s “t” test name?
Would you mind informing the city and country name of the software used in the statistical calculations?
The legend for table 1 needs to be placed on top of the table and not below it.
Generally, tables are “open”; please check the newspaper configuration.
Even if the value of P is not significant, it should be reported in table 1.
As the application route of the two drugs is different (subcutaneous or intramuscular), how did the authors treat this fact? Wouldn’t it be a significant source of bias? Since the application route is different, why didn’t the authors analyze the patients who received each drug (octreotide or lanreotide) separately?
Author Response
Thank you very much for your comments concerning the article „ Self-administration of long-acting somatostatin analogues in NET patients – does it affect the clinical outcome?”. Following your suggestions we corrected the text closely to your comments.
- The Keywords were updated according to the Medical Subject Headings (MeSH).
-
The missing information about the software used for the statistical analysis were added.
-
The legend for table 1 was shifted on the top of the table.
-
The exact P values were added to the table.
-
The information about progression on both long acting somatostatin analogs were added to the result and Table 1.

Reviewer 3 Report
In the final part of the discussion, where you talk about NET progression, it might be useful to compare the presence and possibly the levels of uptake of labeled octreotide with disease progression. In fact, in the presentation of the patients there is a hint of this method which, perhaps, could be better explained
Author Response
Thank you very much for your comments concerning the article „ Self-administration of long-acting somatostatin analogues in NET patients – does it affect the clinical outcome?”.
All patients qualified to the analysis (among all qualified to Somatostatin Analogue treatment) had good or very good (Kenning score 3 and 4) of somatostatin expression on somatostatin receptor imaging (SRI) so the uptake of the somatostatin analogue in imaging does not appear to affect the treatment outcomes. Additionally, for imaging (SSTR expression assessment), we used different somatostatin analogues (TOC and TATE) in different techniques (SPECT / CT and PET / CCT, which would make the analysis not very credible.

Round 2
Reviewer 1 Report
The authors addressed the comments.